# Monitoring the Capacity of *Microsporidia MB* Transgenerational Spread in *Anopheles arabiensis* Populations

**DOI:** 10.3390/insects16121206

**Published:** 2025-11-27

**Authors:** Godfred Yaw Boanyah, Lizette L. Koekemoer, Jeremy K. Herren, Tullu Bukhari

**Affiliations:** 1International Centre of Insect Physiology and Ecology, Nairobi P.O. Box 30772-00100, Kenya; goboanyah@gmail.com; 2Wits Research Institute for Malaria, Faculty of Health Sciences, University of the Witwatersrand, Private Bag 3, Johannesburg 2050, South Africa; lizette.koekemoer@wits.ac.za; 3Centre for Emerging Zoonotic & Parasitic Diseases, Division of the National Health Laboratory Service, National Institute for Communicable Diseases, Johannesburg 2192, South Africa

**Keywords:** *Microsporidia MB*, transgenerational spread, mosquito population, life history traits

## Abstract

*Microsporidia MB* is a natural symbiont of *Anopheles arabiensis* that blocks *Plasmodium* transmission without fitness cost to the mosquito host. Understanding the intergenerational spread of this symbiont in mosquito populations is necessary before applying it in the field as a malaria control tool. The results provide the foundation for improving and maintaining high intensity and prevalence of *Microsporidia MB* in mosquito populations.

## 1. Introduction

*Microsporidia MB*, a recently discovered natural symbiont of *Anopheles* mosquitoes, has been shown to block *Plasmodium* transmission, making it a promising candidate for novel malaria control strategies [1,2]. *Microsporidia MB* is transmitted vertically, from mother to offspring, and horizontally, through mating [2,3,4,5]. Vertical transmission in general is moderate to high. Field-collected *Anopheles arabiensis* Patton, 1905 mothers transmit *Microsporidia MB* to 45–100% of their offsprings on average [2]. Similarly in the sibling species, *An. gambiae* Giles, 1902 sensu stricto (s.s.), vertical transmission has been reported to be 28.6–85.3% in pooled lines but when observed in iso-female lines, vertical transmission can range from 0 to 100% [4]. To date, mating is the only reported mechanism of horizontal *Microsporidia MB* transmission. Horizontal transmission, although heterogeneous, is moderate and higher from infected males to uninfected females (41.5%) than from infected females to uninfected males (22.4%) in *An. arabiensis* in individuals where mating is confirmed [3,5]. *Microsporidia MB*-infected males show twice the mating rate and mating competitiveness compared to uninfected males [3]. Other modes of horizontal transmission, such as ingestion of spores, may also be possible but have not been reported to date. *Microsporidia MB*’s efficient vertical and horizontal transmission likely contributes to its widespread presence in natural *Anopheles* populations across Kenya, Ghana, Niger, and Nigeria. While temperature and humidity influence *Microsporidia MB*’s prevalence and intensity, the symbiont persists across diverse ecological conditions [2,6,7,8,9].

*Microsporidia MB*, an obligate intracellular parasite and symbiont, belongs to the highly host-specific microsporidian group, typically infecting only closely related species [10,11,12]. This narrow host range, combined with poorly understood spore germination triggers and tissue-specific adaptations, complicates in vitro cultivation [10]. A *Microsporidia MB*-based malaria control intervention will likely involve either the dissemination of infective environmental spores or the release of *Microsporidia MB*-infected mosquitoes, either male only or both male and female [1]. Both approaches will rely on the ability of *Microsporidia MB* to spread across natural mosquito populations via vertical and horizontal transmission. Therefore, understanding the natural spread of *Microsporidia MB* across generations under close-to-natural conditions is essential to inform the development of the intervention strategy.

*Microsporidia MB* is similar to *Wolbachia*, because both are naturally occurring symbionts that reduce the potential of mosquitoes to transmit diseases and are transmitted vertically. *Wolbachia* have been reported in wild-caught *Aedes albopictus* Skuse, 1894 and *Aedes aegypti* Linnaeus in Hasselquist, 1762 of which the latter is not a natural host [13,14]. An important difference between *Wolbachia* and *Microsporidia MB* is the capacity for *Microsporidia MB* to be sexually transmitted [5]. A study in India sought to determine the transgenerational spread of *Wolbachia* in *Ae. albopictus* where wild-caught mosquitoes from the field were brought to the laboratory and reared from F1 through to F7 [15]. The results showed an overall increase in *Wolbachia* prevalence from F1 to F7 in four districts while a reverse or decreasing trend was observed in the remaining two districts [15]. These difference in prevalence could be attributed to strain diversity [16].

In addition, environmental factors substantially affect both symbiont transmission and the host in various ways [17,18,19,20]. The diet constitutes a significant aspect of the host–symbiont relationship [21,22]. Moreover, factors such as temperature and humidity have been identified as crucial determinants of the stability and intensity of the symbionts [18,19,20,23]. A study conducted by Doremus et al. [20] on *Cardinium* in the parasitic wasp host, *Encarsia suzannae*, showed that an elevated temperature of 32 °C diminished *Cardinium* intensity, prolonged pupal maturation time, decreased vertical transmission rate, and weakened both cytoplasmic incompatibility modification and rescue efficacy compared to the control at 27 °C. A cool temperature of 20 °C, on the other hand, also diminished symbiont density and extended the host pupal stage, illustrating the substantial influence of temperature on symbionts [20]. Moreover, *Wolbachia* has also been shown to be at high intensity in *D. melanogaster* Meigen, 1830 at 13 °C compared to higher temperatures of 23 °C and 31 °C, respectively [18]. The investigation of the impact of temperature on *Wolbachia* intensity at various developmental stages of the *Ae. albopictus* mosquito demonstrated a notable reduction in bacterial intensity and larval density after exposure to an increased temperature of 37 °C in both sexes [24]. Lastly, a recent study examining four the effect of temperature regimes (22 °C, 27 °C, 32 °C, and 37 °C) on the prevalence and intensity of *Microsporidia MB* in reared mosquito larvae demonstrated a 1.7-fold increase in prevalence and an optimal condition for larval growth at 32 °C [25].

This study aimed to monitor the transgenerational natural *Microsporidia MB* spread under semi-field environmental conditions, as a critical step towards understanding how future field releases will play out in nature. We assessed how *Microsporidia MB* prevalence and infection intensity change across generations and affect mosquito size, while evaluating the influence of temperature and humidity. Understanding these transgenerational effects contributes towards the development of *Microsporidia MB*-based intervention.

## 2. Materials and Methods

### 2.1. Field Collection and Rearing

Wild *Anopheles* mosquitoes (G0) were collected to obtain the first filial generation (F1) required to establish the *Microsporidia MB*-infected mosquito population (Figure 1a). Blood-fed *Anopheles* females resting indoors were collected using aspirators from houses in the villages around the Ahero irrigation scheme (−34.9190 W, −0.1661 N), located in Kisumu County, Kenya. Collections were carried out in two weeks, and the rearing of the F1 to F6 mosquito generations occurred from August 2023 to July 2024. Previous studies show that more than 70% of the anopheline caught in the region belong to the *An. gambiae* complex followed by *An. funestus* (12%) with *An. arabiensis* accounting for over 97% of the *An. gambiae* complex [26]. The mosquitoes collected were transported to the International Centre of Insect Physiology and Ecology (icipe), Thomas Odhiambo Campus (ITOC), Kenya, in cages (30 × 30 × 30 cm^3^) covered with a damp towel, in which mosquitoes had access to a 6% glucose solution. In the laboratory, oviposition was induced by placing individual female mosquitoes (G0) into a 1.5 mL microcentrifuge tube that was lined with filter paper and contained 100 µL of water for moisture [27]. After oviposition, the eggs of each female were placed in larval trays (21 × 15 × 8.5 cm^3^) containing 1 L water. The trays were maintained under semi-field settings until pupation. The female G0 individuals that oviposited were morphologically identified [28] and the *An. gambiae* complex adults were further identified to species level with PCR assay, as described below [29]. Offsprings of only the females (G0) that were *Microsporidia MB*-positive after PCR screening (Figure 1b) and that were identified as *An. arabiensis* were used for the experiments. The pupae (F1) were collected from the larval trays and transferred to cages for emergence (Figure 1c). The temperature and humidity records were taken daily over successive generations (Figure 1d).

### 2.2. Microsporidia MB-Infected Anopheles arabiensis Mosquito Populations

To study the transgenerational spread of *Microsporidia MB*, *Microsporidia MB*-infected *An. arabiensis* mosquito populations were established. *Microsporidia MB* transmission rate from G0 to F1 is between 45% and 100% [2]. Therefore, the F1 populations are expected to be established with somewhere starting from 45% or higher [30]. The purpose of the experiment is then to see how this initial percentage changes across successive generations. A holding cage (30 × 30 × 30 cm^3^) was stocked with 200 F1 adult mosquitoes derived from wild-caught, *Microsporidia MB*-infected *An. arabiensis* (G0). Adult mosquitoes were maintained on a 6% glucose solution provided in vials lined with tissue paper, with small cups containing pupae and glass bowls for oviposition placed on the cage floor. Females were blood-fed via arm feeding (this was done by placing a bare human arm in the cage with mosquitoes for 20 min) five days per week (Figure 1e). Three days after the initial blood feeding, glass oviposition cups lined with filter paper were introduced in the cages. Eggs deposited on the filter paper were transferred to larval trays containing 1 L of water and placed in a screen house. The hatched larvae were fed on Tetramin at 0.3 mg/larva. The pupae from the larval trays were picked daily and transferred to a cage. Each tray and cage were labelled according to the filial generation (F1, F2, F3, etc.) to keep a track of the generation.

Dead mosquitoes were collected daily from each cage and preserved individually in 1.5 mL microcentrifuge tubes for *Microsporidia MB* screening to assess prevalence and intensity across generations. The only exception was the F1 generation in which dead mosquitoes were removed daily starting from the day oviposition was recorded in the cage (this was a precautionary measure as previous attempts had failed). Data loggers were placed in the semi-screened house (or insectary with uncontrolled temperature and humidity) to measure the temperature and humidity as they varied with the natural temperature and humidity levels. Additionally, for each generation, the left wing of 10 males and 10 females (*n* = 20) was excised and measured using a Dino-Lite handheld microscope as a proxy for body size, a parameter of developmental and reproductive fitness. Although seven biological replicates’ populations were initiated, only the replicates that successfully progressed beyond the F2 generation were included in the data analysis.

### 2.3. Deoxyribonucleic Acid (DNA) Extraction and Molecular Species Identification

DNA extraction was done using the ammonium acetate protein precipitation technique [2]. Furthermore, a method described by Santolamazza et al. [29] was used to confirm species identification. Mosquito species were confirmed using a molecular assay that differentiates *An. gambiae s.s.* and *An. arabiensis* using the SINE S200 X6.1 locus (was conducted only on G0 female samples).

### 2.4. Microsporidia MB Screening and Intensity Determination

Quantitative PCR (qPCR) was performed using DNA extracted from mosquitoes to determine if they were infected with *Microsporidia MB* using specific primers (MB18SF: 5′-CGCCGG CCGTGAAAAATTTA-3′ and MB18SR: 5′-CCTTGGACGTG GGAGCTATC-3′) that target the *Microsporidia MB* 18S rRNA region [2]. The PCR detection process volume was 10 µL. This solution included 2 µL of HOT FIREPol Blend Master mix Ready-To-Load (Solis Biodyne, Estonia with Catalogue number: 04-27-00115), 0.5 µL of forward and reverse primers at a concentration of 5 pmol/µL, 2 µL of the DNA template, and 5 µL of nuclease-free water. The thermocycling protocol consisted of an initial denaturation step at 95 °C for 15 min, followed by 35 cycles of denaturation at 95 °C for 1 min, annealing at 62 °C for 90 s, and extension at 72 °C for 60 s. The final auto-extension was performed at a temperature of 72 °C for 5 min. *Microsporidia MB*-positive samples underwent relative qPCR analysis to measure infection levels [2]. The presence of *Microsporidia MB* in each sample was confirmed with a distinctive melt curve related to the *Microsporidia MB* MB18SF/MB18SR primers. The qPCR utilised the MB18SF/MB18SR primers, with normalisation performed using the *Anopheles* ribosomal protein S7 gene (primers, S7F: 5′-TCCTGGAGCTGGAGATGAAC-3′ and S7R: 5′-GACGGGTCTGTACCTTCTGG-3′) as the reference gene. The qPCR was performed using a MIC qPCR cycler (BioMolecular Systems, Upper Coomera, Australia). The qPCR technique was employed to determine both the prevalence and intensity of *Microsporidia MB* in the experimental mosquitoes from F1 to F6 (Figure 1f,g). The following controls were used: no template control (NTC), one known *Microsporidia MB* infected (positive control), and double distilled water (negative control).

### 2.5. Data Analysis

Data for replicate 1 and replicate 3 were significantly different and, therefore, were analysed separately. Sex ratio across generation was compared using a Chi-square test. Regression analysis was conducted to determine the correlation between *Microsporidia MB* prevalence, average temperature, and average humidity. A Friedman test was used to compare *Microsporidia MB* intensities across the generations because the data did not follow a normal distribution. A Wilcoxon signed-rank test was used for pairwise comparisons between the generations with a Bonferroni correction to adjust the *p*-values to avoid risk of a Type I error. A Wilcoxon signed-rank test was also used to compare the *Microsporidia MB* intensities between males and females. Wing size of male and female mosquitoes was compared across generations using a one way Analysis of Variance (ANOVA) followed by pairwise comparisons between the generations with a Bonferroni correction. All analysis were performed in SPSS version 27.

## 3. Results

### 3.1. Overview of Microsporidia MB Mosquito Generations

Two replicates progressed beyond the second generation (F2). Replicate 1 showed stability by the sixth generation with a total of 574 adult individuals (Table 1). Replicate 3, on the other hand, showed a gradual decline with only 28 adults remaining by the sixth generation. The decrease of adults from replicate 3, between F3 (*n* = 281) and F4 (*n* = 72), was due to high adult mortality before oviposition (Table 1). Sex ratio was not significantly different in replicate 1 (χ^2^ = 4.9, df = 4, *p* = 0.3 across generation F2 to F6 and replicate 3 (χ^2^ = 1.43, df = 3, *p* = 0.69) across F2 and F5. The number of males and females from the first generation was not included, as the sex of mosquitoes that died before oviposition was not recorded. Also, under replicate 3, sex was only recorded for the 8 *Microsporidia MB*-positive adult mosquitoes out of the total of 28 at the 6th generation.

### 3.2. Microsporidia MB Prevalence Across Generations and Impact of Temperature and Humidity

*Microsporidia MB* prevalence was determined to predict the prevalence of the symbiont in mosquito populations across generations (Figure 2). For replicate 1, the *Microsporidia MB* prevalence in the mosquito population increased from F1 (63.3%) to F4 (88.3%). The *Microsporidia MB* prevalence for replicate 3 also showed a similar trend with an increase from F1 (73.6%) to F2 (91.5%) (Figure 2). However, *Microsporidia MB* prevalence declined in both replicates, with 30% or less *Microsporidia MB* prevalence by the 6th generation. Sex was not associated with being infected in replicate 1 (χ^2^ = 0.00, df = 1, *p* = 0.97) or replicate 3 (χ^2^ = 0.84, df = 1, *p* = 0.77).

In replicate 1, *Microsporidia MB* prevalence declined after the F4 generation, whereas in replicate 3, the decline began earlier, after the second generation (Figure 2). Regression analysis revealed a significant positive correlation between *Microsporidia MB* prevalence and average temperature (Appendix A) during the adult stage for each generation (R = 0.74, F = 11.83, *p* < 0.01) (Figure 3). In contrast, average humidity during the adult stage for each generation showed no significant effect on prevalence (R = 0.31, F = 1.06, *p* = 0.32) (Figure 3).

### 3.3. Microsporidia MB Intensity Across Generations and Sex

Like the prevalence, *Microsporidia MB* intensity increased, followed by a decrease across generations (Figure 4 and Appendix A). However, unlike *Microsporidia MB* prevalence, regression analysis revealed that both average temperature (R = 0.02, F = 0.85, *p* = 0.35) and average humidity during the adult stage for each generation showed no significant effect on *Microsporidia MB* intensity (R = 0.30, F = 1.90, *p* = 0.17) for both replicates 1 and 3.

A Wilcoxon signed-rank test showed that *Microsporidia MB* intensity was statistically significant between females and males in replicate 1 (Z = −10.50, *p* < 0.00) and replicate 3 (Z = −5.65, *p* < 0.00) (Figure 5). In replicate 1, the median *Microsporidia MB* intensity of females was 5.20 and for male was 0.34. In replicate 3, the *Microsporidia MB* median intensity of females was 0.81 and for males it was 0.06.

### 3.4. Wing Size of Mosquitoes Across Generations

Wing length was used as a proxy for mosquito size across the generations. Wing length was only measured for replicate 1. Replicate 2 was excluded because the last generations failed to meet the minimum sample size for males (10) and females (10). In females, there was an overall significant difference in the wing length across the generations (*F* = 5.69, *df* = 5, *p* < 0.00). Pairwise comparison indicated a difference between the wing length of F1 generation and F3, F4, and F6, with the F1 generation having larger wings (Figure 6 and Appendix A). However, the female wing length of mosquitoes remained similar across generations after the F2 generation. In the case of males, an overall difference was detected (*F* = 2.50, *df* = 5, *p* > 0.04) but pairwise comparison with a Bonferroni correction did not highlight any two generations to be significantly different, indicating that the differences between groups may be too small but collectively cause a significant effect.

## 4. Discussion

This study monitored the spread of *Microsporidia MB* across generations in infected mosquito populations under semi-field conditions. We investigated how the symbiont’s prevalence and intensity are maintained across the early generations, which go through the natural bottleneck effect and revealed a link between environmental temperature and *Microsporidia MB* population prevalence. Additionally, we assessed the effect of these factors on mosquito body size, a proxy of both development and reproductive fitness.

*Microsporidia MB* prevalence and intensity increased in the earlier generations followed by a decrease in subsequent generations across the two surviving biological replicates. Each replicate started with offsprings of only mothers that were infected with *Microsporidia MB*, with no subsequent selection, allowing all the offsprings to contribute towards the following generation. While *Microsporidia MB* may be transmitted both vertically and horizontally, these transmission routes are not 100% efficient [5]. This could be attributed to the localisation of *Microsporidia MB* in the gonads where it is transferred from the germ line cells to the developing eggs [30,31]. In horizontal transmission the intensity of *Microsporidia MB* in the donor female was positively associated with the intensity in the recipient male after mating [3]. It is therefore likely that a decrease in *Microsporidia MB* intensity in a population will result in a decrease in population prevalence of *Microsporidia MB*.

An understanding of the factors that cause the reduction in *Microsporidia MB* prevalence and intensity is essential to optimise the symbiont infection in the mosquito population. The reported infected mosquitoes were established in semi-screened houses with ambient climatic conditions. While humidity has no impact on *Microsporidia MB* prevalence and intensity due to the minimal average changes in humidity over the experimental period, average temperature was positively associated with *Microsporidia MB* prevalence. In nature, however, higher *Microsporidia MB* prevalence has been associated with rainfall in *An. arabiensis* adults [2]. *Microsporidia MB*-infected larvae reared under a high average temperature of 32 °C were found to develop in a shorter time and have a higher infection rate compared to infected larvae reared under lower temperatures [25]. This study shows that the average temperature during the adult stage also contributes to higher infection rates. Subsequent generations of *Wolbachia*-infected mosquitoes have also been shown to be influenced by temperatures that are higher than 28 °C [32], in addition to a significant number of other symbiotic microorganisms [33]. It is noteworthy that, in addition to average temperature, other environmental factors such as water quality may also influence *Microsporidia MB* infection.

In both replicates, female mosquitoes had higher infection intensity compared to males. Higher *Microsporidia MB* intensity in females has also been reported previously when both infected males and females were reared on similar larval and adult diets [21]. In this study, the females were blood-fed which could have provided resources that *Microsporidia MB* exploits for replication, leading to higher intensity in females. Significantly higher relative intensities of *Microsporidia MB* are found in the gonads of blood-fed *An. arabiensis* compared to sugar-fed mosquitoes [31]. This may partially explain the higher *Microsporidia MB* intensity in females. Other factors to consider include the possibly lower evolutionary pressure for *Microsporidia MB* to infect males due to lower contribution to vertical transmission, shorter lifespans of males compared to females and therefore less time for *Microsporidia MB* to proliferate, variation in immune response to *Microsporidia MB* infection, or loss of heavy loads of *Microsporidia MB* following mating.

In addition to *Microsporidia MB* intensity, females also varied from males in how colonisation impacted their body size. While male body size remained similar across generations, the females body size was significantly smaller in the second generation, beyond which it did not change. Body size is crucial as it is a measure of both developmental and reproductive fitness [34,35,36,37]. Conveniently, body size can be maintained by adjusting the quantity and quality of diet available [21,22,38]. Noting that this result is based on a small sample from one replicate, a follow up study on adjusting the diet of female mosquitoes especially in the early generations and determining its outcome on *Microsporidia MB* transmission to the following will be helpful.

This study successfully determined the intergenerational spread of *Microsporidia MB* in *Anopheles arabiensis* under semi-field conditions, demonstrating its potential for malaria control while revealing key challenges in maintaining symbiont prevalence and intensity across generations. Future research should focus on optimizing environmental conditions such as temperature modulation and selective breeding of high-intensity individuals to enhance vertical and horizontal transmission rates for long-term maintenance of *Microsporidia MB* in *Anopheles* population. Additionally, studies on an improved diet for female mosquitoes and advancement in developing artificial infection methods (e.g., spore inoculation or microinjection) will support this too. These findings together with further development will be critical for scaling up production and implementing *Microsporidia MB*-based malaria transmission blocking strategy.

## Figures and Tables

**Figure 1 insects-16-01206-f001:**
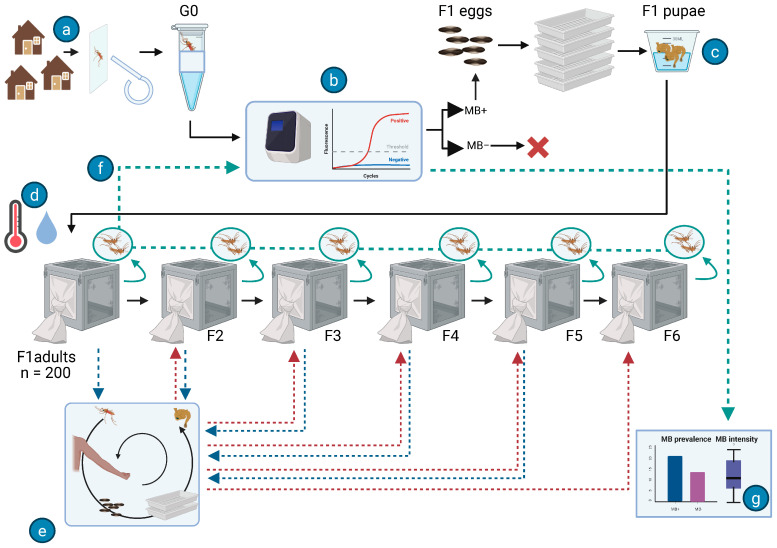
The schematic diagram showing the (**a**) collection from houses, (**b**) screening of *Microsporidia MB*-infected G0, and (**c**) rearing of F1 generation from larvae to pupa stage. The pupae were transferred to cages kept in an insectary where (**d**) temperature and humidity were recorded daily. The female mosquitoes in the cage (blue dotted arrow) were (**e**) arm-fed and the eggs that were laid were reared in larval trays. The resulting pupae were transferred to the cage designated for the next generation (maroon dotted arrow). (**f**) Dead mosquitoes were picked daily from each cage and screened for *Microsporidia MB* (green dotted line) to determine *Microsporidia MB* prevalence and intensity per generation (**g**).

**Figure 2 insects-16-01206-f002:**
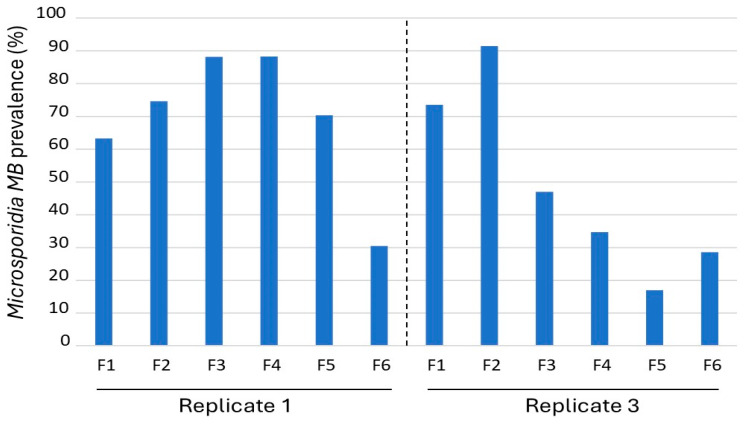
Prevalence of *Microsporidia MB* across generations in two replicates of *An. arabiensis* mosquito population.

**Figure 3 insects-16-01206-f003:**
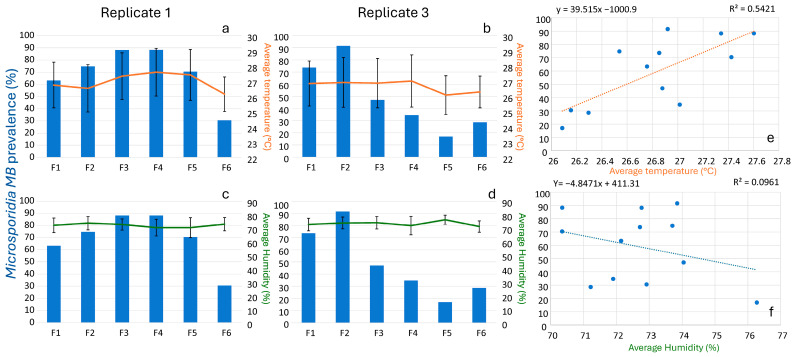
Prevalence of *Microsporidia MB* across generations in relation to average temperature (°C) ± S.D. ((**a**,**b**) for replicate 1 and 3 respectively) and humidity (%) ± S.D. ((**c**,**d**) for replicate 1 and 3 respectively) during the adult stage of the mosquitoes. Also shown is the correlation between the *Microsporidia MB* prevalence across generations in both replicates, (**e**) average temperature, and (**f**) average humidity during the adult stage of *An. arabiensis* mosquitoes.

**Figure 4 insects-16-01206-f004:**
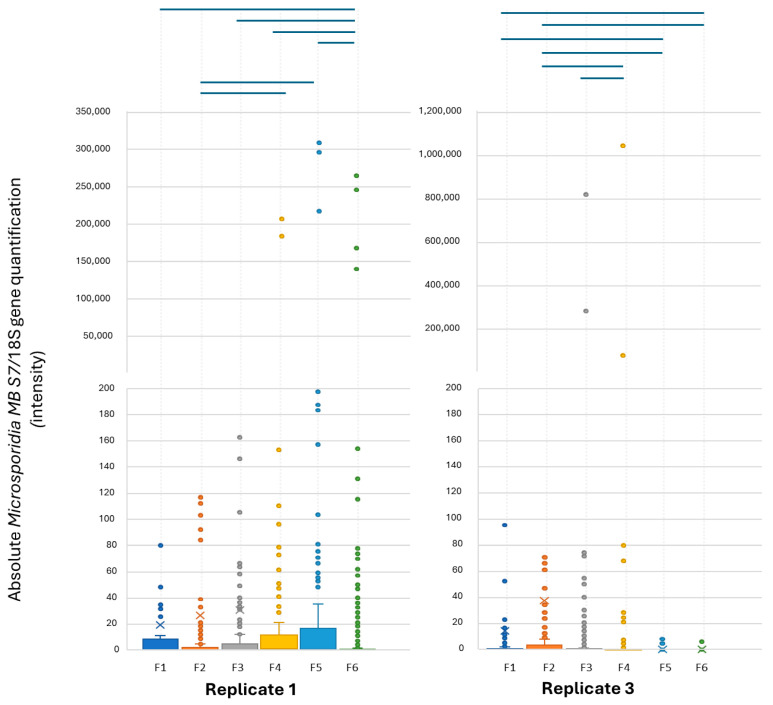
Box plots indicating the median relative *Microsporidia MB* intensities across generations in replicate 1 and replicate 3. Whiskers represent the IQR and dots represent the outliers. Blue lines indicate a significant difference between the generations after adjustment for multiple comparison (Bonferroni correction).

**Figure 5 insects-16-01206-f005:**
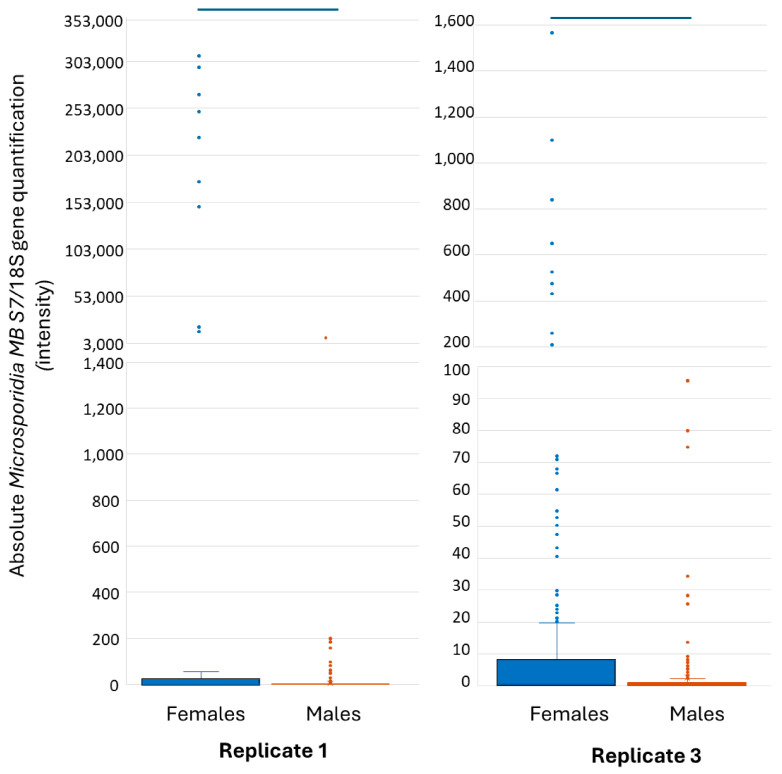
Box plots indicating the median relative *Microsporidia MB* intensities for females and males combined across the generations in replicate 1 and replicate 3. Whiskers represent the IQR and dots represent the outliers. Blue lines indicate a significant difference between females and males.

**Figure 6 insects-16-01206-f006:**
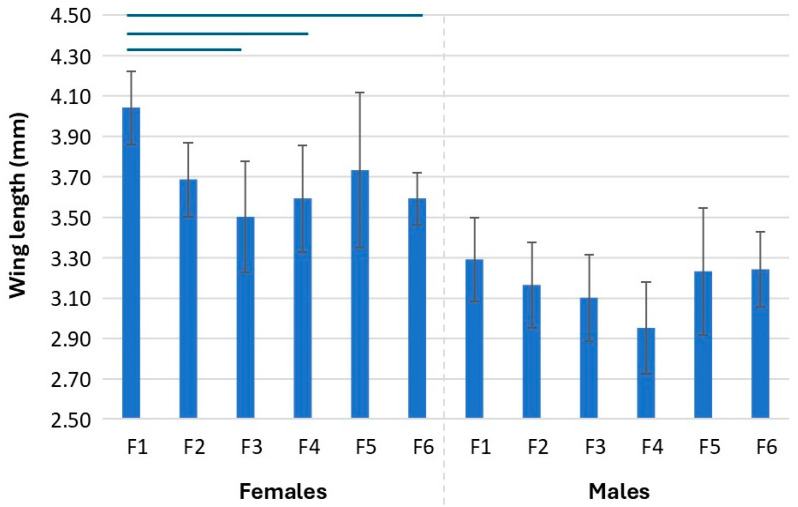
Female and male wing size (mm (S.D.)) across generations for replicate 1. Lines on top indicate a significant difference between the generations for females. No significant difference was found in wing length between the generations for males.

**Table 1 insects-16-01206-t001:** Frequency table of adult *Microsporidia MB* mosquitoes (males and females) across generations in replicate 1 and 3.

	Replicate 1			Replicate 3		
Generation (F)	Female	Male	Total	Female	Male	Total
**1 ***	50	10	60	84	98	182
**2**	84	106	190	81	72	153
**3**	153	178	331	162	119	281
**4**	61	55	116	41	31	72
**5**	99	101	200	27	26	53
**6**	296	278	574	3 **	5 **	28

Each replicate was started with 200 F1 adult progeny of *Microsporidia MB*-infected *Anopheles arabiensis*. * Represents the number out of the starting 200 that was in the cage when the first oviposition was recorded. ** Sex was recorded for only 8 out of the 28 samples. These eight samples were all the *Microsporidia MB* infected samples in the generation.

## Data Availability

The original data for this study are included in the article/Appendix A. Further inquiries can be directed to the corresponding authors.

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
