# Peer review of "Monitoring the Capacity of Microsporidia MB Transgenerational Spread in Anopheles arabiensis Populations"

_insects, 2025, doi:10.3390/insects16121206_

Round 1
Reviewer 1 Report
Comments and Suggestions for Authors
I read the manuscript with interest. The topic is certainly of interest both for its aspects of symbiont-host coevolution and for its application in the fight against vector-borne diseases.
The work is well-structured, the methodologies well-described and consistent with the objectives.
It is certainly an excellent work, for which I suggest some minor changes.
Introduction: The introduction discusses the characteristics of Wolbachia and Microsporidia. Then, a reference to Cardinium-wasp discusses cytoplasmic incompatibility. I believe this aspect should be brought forward and referred to the characteristics of Wolbachia, otherwise it seems out of context.
Typos:
Line 77: Ae. albopictus should be corrected to Ae. albopictus
Line 164: An. arabiensis should be corrected to An. arabiensis
Line 133: Santomaza should be corrected to Santolamazza
Line 195: n insectary should be corrected to in insectary
Author Response
Reviewer 1
It is certainly an excellent work, for which I suggest some minor changes.
- Introduction: The introduction discusses the characteristics of Wolbachia and Microsporidia. Then, a reference to Cardinium-wasp discusses cytoplasmic incompatibility. I believe this aspect should be brought forward and referred to the characteristics of Wolbachia, otherwise it seems out of context.
We thank the reviewer for this observation. The mention of Cardinium and its effect on cytoplasmic incompatibility was included to illustrate how environmental factors, particularly temperature, can influence symbiont transmission and associated reproductive phenotypes. While Microsporidia MB does not induce cytoplasmic incompatibility, this example highlights the broader principle that environmental variation can affect symbiont stability, density, and transmission efficiency across taxa. Therefore, we prefer to retain the current structure as it effectively demonstrates the general influence of environmental conditions on symbiont–host interactions.
Typos:
Line 77: Ae. albopictus should be corrected to Ae. albopictus
Line 164: An. arabiensis should be corrected to An. arabiensis
Line 133: Santomaza should be corrected to Santolamazza
Line 195: n insectary should be corrected to in insectary
Response: Thanks very much for this valuable comment. We have corrected the typo errors as requested in the manuscript.
Reviewer 2 Report
Comments and Suggestions for Authors
Dear Authors,
The work you have done is very interesting and relevant for science but also for public health, particularly to the public health of the African continent. Design of the study is mainly clear but there are some issues that are not understandable. Before published, this manuscript requires revision due to the unclear parts in Materials and Methods. Also, the photos of the experiment should be added. For such interesting research, the photos are really needed.
Please find below my comments and suggestions:
L25 The sentence “All mosquitoes survived till death” has to me modified because we all survive until death.
L45 Please add the author and the year when species is mentioned first time in the text.
L72 The part od sentence “in that both are…” should be simplified to “because both are…”
L73 Please correct to…that reduce …
L74 Add year and author when species is mention for the first time in text. Act accordingly in the whole manuscript.
L77 Please correct Ae. Al. It was auto-corrected.
L84 Citation 17 is not good. The cited study is not about transmission.
L87 Please delete coma before bracket.
L94 The name should be shorten D. melanogaster.
L129 Correct (Figure 1: c). to have the same format as others.
In the text is 1a then 1c and then 1b. It should have logical order in the text.
L133 Please delete coma and add space: Santomaza et al.,[29] was
L164 There is auto-correct
L169 Two week? You said above that collection period was from August 2023 to July 2024. Please clarify.
L172 Please explain more in details blood-feeding.
L182 Why was different in F1?
L189 were included
It is not clear here what the replicates were. Please make that very important part clear. How many replicates were performed?
L211-214 Please simplify sentence. It is to long and hard to understand.
Table 1. The title should be more explanatory because it is not clear what was done. You said that in F1 was 200 and then you said in title that it is the number of adult mosquitoes. Please simplify it and add more details.
L239 Instead of ! it is more common to use **
Table 1. Correct Fe-male to female in the same line.
Figure 2. Numbers should be added to all graphs.
The authors showed statistic analyses but it is not clear what the replicates were except these two. With two numbers we cannot perform statistics. Please explain what was done. Is it every mosquito one repletion? Or?
Please correct numbers to two decimals. It is enough.
Figure 3. It is not clear what is a-c or b-d. Please explain.
Why the wing length was not measured for all available specimens? Why only replicate 1? Second issue is that there is possibility that with generations wings become smaller due to its development in the lab. The field specimens and F1 are the closest morphologically because the lab rearing did not do much in the selection yet. Please reconsider if this should stay.
L323-324 Not clear sentence because of increase and decrease. Please simplify.
L346 Please add the ℃.
Author Response
Reviewer 2
Please find below my comments and suggestions:
- L25 The sentence “All mosquitoes survived till death” has to me modified because we all survive until death.
Response: Thank you, we agree. The phrase “All mosquitoes survived till death” has been rephrased to “All the mosquitoes… after death”.
- L45 Please add the author and the year when species is mentioned first time in the text.
Response: Thanks very much. I have included the author and year for the first time mention of species in the text eg. Anopheles arabiensis Patton, 1905, Aedes aegypti Linnaeus in Hasselquist, 1762, Aedes albopictus Skuse, 1894 and An. gambiae Giles, 1902 sensu stricto (s.s.).
- L72 The part of sentence “in that both are…” should be simplified to “because both are…”
Response: I have simplified “in that both are…” to “because both are…” in the manuscript. Thank you
- L73 Please correct to…that reduce …
Response: ” that reduces” has been corrected to “that reduce”
- L74 Add year and author when species is mention for the first time in text. Act accordingly in the whole manuscript.
Response: Author and year have been included in first mention of species throughout the manuscript
- L77 Please correct Ae. Al. It was auto-corrected.
Response: Thanks very much the auto-correct has been corrected from Ae Albopictus to Ae. albopictus
- L84 Citation 17 is not good. The cited study is not about transmission.
We thank the reviewer for this comment. The intent of the sentence was to highlight that environmental factors influence both the symbiont and its insect host. Therefore, citation 17 was included to support the aspect of environmental effects on the host insect, which is relevant to the overall context of host–symbiont interactions discussed in this section. We have thus retained the citation as originally placed.
- L87 Please delete coma before bracket.
Response: The coma before bracket has been deleted
- L94 The name should be shorten D. melanogaster.
Response: Drosophila melanogaster has been shortened to D. melanogaster
- L129 Correct (Figure 1: c). to have the same format as others. In the text is 1a then 1c and then 1b. It should have logical order in the text.
Response: Thanks very much, Figure 1: c has been corrected to Figure 1c just like the others. Also, the text has been rearranged to allow the Figure numbering to follow a logical order as requested.
- L133 Please delete coma and add space: Santomaza et al.,[29] was
Response: The coma has been deleted and space was added hence it now reads “Santolamazza et al. [29] was”
- L164 There is auto-correct
Response: The auto-correct has been corrected to An. arabiensis
- L169 Two week? You said above that collection period was from August 2023 to July 2024. Please clarify.
Response: The two weeks were for field collection of the G0 while August 2023 to July 2024 was for the duration of rearing the replicates from F1 to F6. I have clarified in the manuscript, and it now reads “Collections were carried out in two weeks, and the rearing of F1 to F6 mosquito generations occurred from August 2023 to July 2024.”
- L172 Please explain more in details blood-feeding.
Response: The following statement has been added to explain how blood-feeding was done “This was done by placing a bare human arm in cage with mosquitoes for 20 mins”
- L182 Why was different in F1?
Response: This was a precautionary measure as previous three attempts failed so in order to save time record of dead mosquitoes for F1 was made only after oviposition
- L189 were included
It is not clear here what the replicates were. Please make that very important part clear. How many replicates were performed?
Response: Biological replicates have been included in the sentence for clarity. There were seven attempts in all with three collapsing in F1 and one in F2 while the two used in the analysis progressed beyond F2.
- L211-214 Please simplify sentence. It is too long and hard to understand.
Response: Thanks very much. The sentence has been simplified to “Friedman test was used to compare Microsporidia MB intensities across the generations because the data did not follow a normal distribution”
- Table 1. The title should be more explanatory because it is not clear what was done. You said that in F1 was 200 and then you said in title that it is the number of adult mosquitoes. Please simplify it and add more details.
Response: The title for Table 1 has been rephrased to “Frequency table of adult Microsporidia MB mosquitoes (males and females) across the generations in replicate 1 and 3”. Details have been added below the table.
- L239 Instead of ! it is more common to use **
Response: Thank you, “!” has been replaced with “**”
- Table 1. Correct Fe-male to female in the same line.
Response: “Fe-male” to “Female” in the same line as requested.
- Figure 2. Numbers should be added to all graphs.
The authors showed statistic analyses but it is not clear what the replicates were except these two. With two numbers we cannot perform statistics. Please explain what was done. Is it every mosquito one repletion? Or?
Please correct numbers to two decimals. It is enough.
Response: The number of mosquitoes are provided in Table 1 and the prevalence of Microsporidia MB (%) is what the bar indicates in figure 2. Statistical analysis was carried out to compare the prevalence across the six generations and between male and female mosquitoes. The numbers have been corrected to two decimals.
- Figure 3. It is not clear what is a-c or b-d. Please explain.
Why the wing length was not measured for all available specimens? Why only replicate 1? Second issue is that there is possibility that with generations wings become smaller due to its development in the lab. The field specimens and F1 are the closest morphologically because the lab rearing did not do much in the selection yet. Please reconsider if this should stay.
Response: The title of Figure 3 has been rephrased for clarity and now reads “Prevalence of Microsporidia MB across generations in relation to average temperature (°C) ± S.D. (a and b for replicate 1 and 3 respectively) and humidity (%) ± S.D. (c and d for replicate 1 and 3 respectively) during the adult stage of the mosquitoes. Also shown is the correlation between the Microsporidia MB prevalence across generations in both replicates and (e) average temperature (f) average humidity during the adult stage of An. arabiensis mosquitoes”.
The wing length of replicate 3 was not measured and included in the analysis because F6 under replicate 3 failed to meet the minimum sample size of 10 males and 10 females. Hence the reason why only replicate one was used for the analysis.
We thank the reviewer for this valuable comment and agree with the observation. The measurement of wing length was included because it serves as a reliable proxy for body size, which can influence reproductive and developmental fitness parameters. Our aim was to confirm that body size was not a confounding factor affecting offspring number, mating rates, or consequently, the vertical and horizontal transmission of Microsporidia MB. We did not observe any overall difference in female wing length across generations. In males, although variation was noted, no two generations were significantly different. It is acknowledged that in newly established laboratory colonies, a reduction in body size can occur when rearing densities are high or procedures are not standardized; however, this was not the case in our study.
L323-324 Not clear sentence because of increase and decrease. Please simplify.
Response: Thank you, the sentence has been simplified to “Microsporidia MB prevalence and intensity increased in the earlier generations followed by a decrease in subsequent generations across the two surviving biological replicates”
- L346 Please add the ℃.
Response: Thank you, “degree Celsius” has been replaced with “℃”
Round 2
Reviewer 2 Report
Comments and Suggestions for Authors
The Authors corrected all that was suggested and replied to all comments. Therefore, I suggest acceptance of the MS.